# Diagnostic Performance of Various Ultrasound Risk Stratification Systems for Benign and Malignant Thyroid Nodules: A Meta-Analysis

**DOI:** 10.3390/cancers15020424

**Published:** 2023-01-09

**Authors:** Ji-Sun Kim, Byung Guk Kim, Gulnaz Stybayeva, Se Hwan Hwang

**Affiliations:** 1Department of Otolaryngology-Head and Neck Surgery, Eunpyeong St. Mary’s Hospital, College of Medicine, The Catholic University of Korea, Seoul 06591, Republic of Korea; 2Department of Physiology and Biomedical Engineering, Mayo Clinic, Rochester, MN 55902, USA; 3Department of Otolaryngology-Head and Neck Surgery, Bucheon St. Mary’s Hospital, College of Medicine, The Catholic University of Korea, Seoul 06591, Republic of Korea

**Keywords:** thyroid cancer, thyroid nodules, ultrasonography, meta-analysis, diagnostic imaging

## Abstract

**Simple Summary:**

In the present study, the sensitivity, specificity, and pooled diagnostic performances according to the cutoff value for diagnosing cancer of five ultrasound risk-stratification systems often used in clinical practice were verified by performing a meta-analysis. Sixty-seven studies involving 76,512 thyroid nodules were included in this research. The highest area under the curve (AUCs) of the K-TIRADS, ACR-TIRADS, ATA classification, EU-TIRADS, and Kwak-TIRADS were 0.904, 0.882, 0.859, 0.843, and 0.929, respectively. Based on the optimal sensitivity and specificity, the AUC or diagnostic odds ratios of K-TIRADS, ACR-TIRADS, ATA, EU-TIRADS, and Kwak-TIRADS were taken as the cutoff values of 4 (intermediate suspicion), TR5 (highly suspicious), high suspicion, 5 (high risk), and 4b, respectively. All ultrasound-based risk-stratification systems had good diagnostic performance.

**Abstract:**

Background: To evaluate the diagnostic performance of ultrasound risk-stratification systems for the discrimination of benign and malignant thyroid nodules and to determine the optimal cutoff values of individual risk-stratification systems. Methods: PubMed, Embase, SCOPUS, Web of Science, and Cochrane library databases were searched up to August 2022. Sensitivity and specificity data were collected along with the characteristics of each study related to ultrasound risk stratification systems. Results: Sixty-seven studies involving 76,512 thyroid nodules were included in this research. The sensitivity, specificity, diagnostic odds ratios, and area under the curves by K-TIRADS (4), ACR-TIRADS (TR5), ATA (high suspicion), EU-TIRADS (5), and Kwak-TIRADS (4b) for malignancy risk stratification of thyroid nodules were 92.5%, 63.5%, 69.8%, 70.6%, and 95.8%, respectively; 62.8%, 89.6%, 87.2%, 83.9%, and 63.8%, respectively; 20.7111, 16.8442, 15.7398, 12.2986, and 38.0578, respectively; and 0.792, 0.882, 0.859, 0.843, and 0.929, respectively. Conclusion: All ultrasound-based risk-stratification systems had good diagnostic performance. Although this study determined the best cutoff values in individual risk-stratification systems based on statistical assessment, clinicians could adjust or alter cutoff values based on the clinical purpose of the ultrasound and the reciprocal changes in sensitivity and specificity.

## 1. Introduction

The thyroid gland is an organ that can be easily inspected by using ultrasound (US). US is an accurate test that can confirm the characteristics of the thyroid and is a highly accessible diagnostic method that can be performed relatively easily in an outpatient setting [1]. Thyroid US is a primary imaging test for the evaluation of thyroid nodules, and the evidence of thyroid cancer has been confirmed through imaging features of thyroid nodules [2]. The popular use of US has increased the diagnostic rate for thyroid nodules [3]. However, this does not mean that the incidence of thyroid cancer or the need for treatment has increased. There was a report that the more US was performed, the more thyroid cancer was diagnosed [4]. Concerns have been raised about unnecessary biopsies and additional tests for benign thyroid nodules or thyroid cancer with infrequent progression. The low mortality and high diagnostic rates have given rise to a discussion of overdiagnosis.

US-based risk stratification systems (RSSs) have been proposed by several international societies to prevent the overdiagnosis of thyroid nodules and to help determine additional tests and follow-up. RSS is being applied in clinical practice as a method of classifying and scoring characteristic findings of thyroid nodules. Even after the first meta-analysis was performed in 2019 [5], many studies reported the diagnostic accuracy of each RSS. In the present study, the sensitivity, specificity, and pooled diagnostic performances according to the cutoff value for diagnosing cancer of five RSSs often used in clinical practice were verified by performing a meta-analysis. In addition, the clinical implications of the diagnostic accuracy were reviewed.

## 2. Materials and Methods

### 2.1. Study Protocol and Literature Search Strategy

This meta-analysis was performed in accordance with the Preferred Reporting Items for Systematic Reviews and Meta-Analyses guidelines [6]. The study protocol was prospectively registered on the Open Science Framework (https://osf.io/7cu2y/ (accessed on 27 September 2022)). Clinical studies were retrieved from PubMed, Embase, SCOPUS, Web of Science, and the Cochrane Central Register of Controlled Trials from the start date to August 2022. The search terms were as follows: thyroid, thyroid nodule, thyroid neoplasm, malignancy, thyroid cancer; diagnostic imaging, diagnostic performance diagnostic value, ultrasonography, diagnosis, ultrasound, diagnostic value, ultrasonography, ultrasound classifications, ultrasound risk stratification system, imaging, reporting systems, thyroid imaging reporting and data system (TI-RADS), TI-RADS, TIRADS, Indeterminate, Korean Society of Thyroid Radiology and Korean Thyroid Association guideline (K-TIRADS), American College of Radiology guideline (ACR-TIRADS), American Thyroid Association (ATA) guidelines, European Thyroid Association guideline (EU-TIRADS), and Kwak-TIRADS (Appendix A). Two independent reviewers removed studies that were not related to the diagnosis or prediction of thyroid malignancy using US classifications by assessments of article titles, abstracts, and full texts.

### 2.2. Selection Criteria

The inclusion criteria were as follows: articles about patients undergoing US of thyroid nodules, and comparison of US findings with cytologic or histologic findings. Exclusion criteria included review articles, case reports, articles about other neck diseases (e.g., lymphadenitis or neck mass), articles without adequate data to determine the diagnostic value of US, and those not written in English.

### 2.3. Data Extraction and Risk of Bias Assessment

Data from articles included in the study were extracted in a standardized format [7]. The results of the analysis were diagnostic odds ratio (DOR), summary receiver operating characteristic (SROC) curve, and area under the curve (AUC). DOR was calculated by using the parameters of true positive, true negative, false positive, and false negative. The DOR was assessed with a 95% confidence interval by using a random effects model. The SROC curve and AUC were used as methods to evaluate diagnostic data in meta-analysis. As the discriminant power of the test increases, the SROC curve gets closer to the upper left corner, the point where both sensitivity and specificity are 100% [8]. Higher AUC values range from zero to one, indicating better test performance. The AUC value indicated diagnostic accuracy [9].

The Quality Assessment of Diagnostic Accuracy Studies Version 2 tool (QADAS-2) was used to evaluate methodological quality (risk of bias) [10]. For the definition of true positive and negative, guideline category < cutoff value was regarded as “test negative” and guideline category ≥ cutoff value as “test positive.” Therefore, “benign” lesions classified as <cutoff were regarded as true negative, and “non-benign” lesions classified as ≥cutoff value were regarded as true positive. Accordingly, the sensitivity, specificity, and DOR were calculated with reference to the results based on pathological examination, or fine-needle aspiration (FNA) cytology and follow-up. Receiver operating characteristic (ROC) curve analyses and areas under the ROC (AUC) were used to assess the value of guidelines in differentiating benign from malignant thyroid nodules.

### 2.4. Statistical Analysis and Outcome Measurements

R statistical software (R Foundation for Statistical Computing, Vienna, Austria) was used for this analysis. To assess heterogeneity, a homogeneity analysis was performed by using the Q statistic. According to the 2016 Korean Society of Thyroid Radiology and Korean Thyroid Association guidelines, thyroid nodules were assigned to be benign, of low suspicion (K-TIRADS 3), intermediate suspicion (K-TIRADS 4), and high suspicion (K-TIRADS 5) [2]. The US features in the ACR TI-RADS are categorized as benign (TR1, 0 point), not suspicious (TR2, 2 points), mildly suspicious (TR3, 3 points), moderately suspicious (TR4, 4–6 points), or highly suspicious (TR5, 7 points or more) for malignancy [11]. Based on the 2015 ATA guidelines, the thyroid nodules were classified according to the malignancy risk as “high”, “intermediate”, “low” or “very low” suspicion [12]. EU-TIRADS classified thyroid nodules as benign and low-, intermediate-, and high-risk nodules according to the malignancy risk [1]. The TI-RADS categories proposed by Kwak et al. classify thyroid nodules as 2 (benign lesions), 3 (no suspicious US features), 4a (one suspicious US feature), 4b (two suspicious US features), 4c (three or four suspicious US features), and 5 (five suspicious US features) according to the risk estimates of malignancy [13]. Diagnostic accuracy in individual risk stratification systems (K-TIRADS, ACR-TIRADS, ATA, EU-TIRADS, and Kwak-TIRADS) was assessed based on the use of different cutoff values. Potential publication bias was assessed by Begg’s funnel plot and Egger’s linear regression test.

## 3. Results

### 3.1. Search and Study Selection

After screening 3148 articles through an established process, a total of 746 articles were excluded after reviewing the relevance of titles and abstracts. A full-text review of the remaining 86 articles was performed, and 19 articles were excluded because they analyzed other interventions or lacked results. As a result, 67 studies with 76,512 thyroid nodules were included in the analysis (Figure 1) [14,15,16,17,18,19,20,21,22,23,24,25,26,27,28,29,30,31,32,33,34,35,36,37,38,39,40,41,42,43,44,45,46,47,48,49,50,51,52,53,54,55,56,57,58,59,60,61,62,63,64,65,66,67,68,69,70,71,72,73,74,75,76,77,78]. The characteristics of the study are presented in Appendix A and the results of the bias assessment are presented in Appendix A. Egger’s test yielded a significant result (*p* > 0.05) except sensitivity of ACR TI-RADS TR3 (*p*-value = 0.009114), K-TIRADS 3 (*p*-value = 0.0001452), Kwak-TIRADS 4a (*p*-value = 0.002071), and Kwak-TIRADS 4b (*p*-value = 0.03186). However, all biased outcomes showed no significance between original and corrected (trim fill method) outcomes. Begg’s funnel plots for each RSS are presented in Appendix A.

### 3.2. Diagnostic Accuracy in Various US Risk Stratification Systems

In K-TIRADS categories, sensitivity changed from around 66% to 99% (highest in low suspicion) and specificity showed an inverse association (89% to 8%; highest in high suspicion) according to the different cutoff values (categories) (Table 1). ROC analysis and DOR showed that the best diagnostic cutoff values of K-TIRADS were low and intermediate suspicion, respectively (Figure 2 and Figure 3). Although a test with high AUC is statistically considered “better” than one with lower AUC, AUC lacks clinical interpretability because it does not reflect the practical gains and losses of individual patients by diagnostic tests. In addition, AUC can consider a test that increases sensitivity at low specificity superior to one that increases sensitivity at high specificity [79]. If the sensitivity and specificity on a screening test were considered to be too high or too low, they could be adjusted by raising or lowering cutoff values [80]. Based on the above mentioned, the best cutoff value of K-TIRADS was category intermediate suspicion (K-TIRADS 4) with the sensitivity of 92.5% and specificity of 62.8%.

In ACR-TIRADS categories, sensitivity changed from around 63.5% to 98.4% (highest in TR3) and specificity showed an inverse association (89.5% to 22.8%; highest in TR5) according to the different cutoff values (categories) (Table 2). ROC analysis and DOR showed that the best diagnostic cutoff values of ACR-TIRADS were TR5 in common (Appendix A). Based on the statistical results, the best cutoff value of ACR-TIRADS was category TR5 with the sensitivity of 63.5% and specificity of 89.5%. However, TR4 would also be a good cutoff value of ACR-TIRADS based on another clinician’s opinion that high sensitivity could be more suitable than high specificity in the screening test.

In ATA categories, sensitivity changed from around 69.7% to 97.6% (highest in low suspicion) and specificity showed an inverse association (87.1% to 22.6%; highest in high suspicion) according to the different cutoff values (categories) (Table 3). ROC analysis and DOR showed that the best diagnostic cutoff values of ATA had high suspicion in common (Appendix A). Statistically, the best cutoff value of ATA was category the “high” with the sensitivity of 69.7% and specificity of 87.1%. However, similar to ACR-TIRADS, intermediate suspicion would also be a good cut-off value of ATA in another clinician’s opinion that high sensitivity could be more suitable than high specificity in the screening test.

In EU-TIRADS categories, sensitivity changed from around 70.6% to 99.1% (highest in low risk) and specificity showed an inverse association (83.9% to 3%; highest in high risk) according to the different cutoff values (categories) (Table 4). ROC analysis and DOR showed that the best diagnostic cutoff values of EU-TIRADS were high risk and intermediate risk, respectively (Appendix A). Statistically, the best cutoff value of EU-TIRADS was high risk with the sensitivity of 70.6% and specificity of 83.9%. However, like ACR-TIRADS, intermediate risk would also be a good cutoff value for EU-TIRADS in another clinician’s opinion that high sensitivity could be more suitable than high specificity as the screening test.

In Kwak-TIRADS categories, sensitivity changed from approximately 15% to 99% (highest in 4a) and specificity showed an inverse association (99% to 32%; highest in 5) according to the different cutoff values (categories) (Table 5). ROC analysis and DOR showed that the best diagnostic cutoff values of Kwak-TIRADS were 4a and 4b, respectively (Appendix A). A cutoff in the screening test has been chosen to minimize the rate of false negatives rather than reducing false positives, because this would be appropriate for conditions in which misdiagnosing and treating someone as sick is better than missing truly sick individuals [81]. Based on practical and statistical considerations, the best cutoff value of Kwak-TIRADS was category 4b with the sensitivity of 95.8% and specificity of 63.7%.

## 4. Discussion

The US-based RSSs have been useful for diagnosing thyroid nodules in clinical practice over the past decade (Appendix A). There have been many previous studies confirming the usefulness of each RSS. However, a comprehensive analysis of the cutoff value for diagnosing thyroid cancer has been lacking. Therefore, we confirmed the diagnostic accuracy and cutoff value including the most recent clinical studies for each RSS. This study analyzed the results of 76,512 thyroid nodules from 67 studies. The highest AUC of the K-TIRADS was 0.904 for low suspicion, but the false positive rate was high with a low specificity of 8%. Based on DOR, intermediate suspicion (K-TIRADS 4) showed the highest diagnostic accuracy. ACR-TIRADS showed the highest accuracy with AUC 0.882 in TR 5 and high sensitivity of 92.5% in TR 4. ATA classification demonstrated the highest diagnostic accuracy with an AUC of 0.859 in high suspicion and a high sensitivity of 88% in intermediate suspicion. In EU-TIRADS, EU-TIRADS 5 showed the highest diagnostic accuracy of 0.843, and EU-TIRADS 4 showed a high sensitivity of 93%. In Kwak-TIRADS, 4b showed a high AUC of 0.925 and sensitivity of 95.8%. All values except low suspicion of ATA classification and EU-TIRADS showed good diagnostic accuracy of more than DOR 10 [82].

When compared with a meta-analysis study of the diagnostic performance of the four RSSs performed in 2019 [5], the sensitivity and AUC of K-TIRADS cutoff values 4 and 5 were higher in the present study. The sensitivity, specificity, and AUC of ACR-TIRADS in TR5 were similar to present study results or slightly higher in the previous meta-analysis. When TR4 was used as the cutoff value, sensitivity and AUC were lowered in the present study (95% vs. 92.5%, 0.88 vs. 0.75, respectively). In ATA and EU-TIRADS, both high and intermediate categories showed lower diagnostic accuracy in the present study, which included an additional 34 studies published after 2020. It was interesting that the sensitivity and diagnostic accuracy of K-TIRADS was higher than those of the meta-analysis performed in 2019, and that many studies on K-TIRADS were added to the present study.

Thyroid nodules are relatively common, and the incidence rate confirmed through palpation is approximately 4%, but after the popularization of US examination, the incidence rate was reported to be as high as 70% [3]. US can screen for thyroid cancer by confirming the imaging characteristics such as composition, echogenicity, shape, and margin of the thyroid nodule [73]. It is also the basis for deciding whether to proceed with additional diagnostic tests such as FNA biopsy and core needle biopsy [83]. According to a population study, the number of patients diagnosed with papillary thyroid cancers (PTCs) increased rapidly at about 3% per year [84]. Meanwhile, mortality from thyroid cancer was found to remain stable, inferring that a large proportion of PTCs is due to the overdiagnosis of low-risk tumors [85,86]. Various US-based RSSs have been proposed to avoid unnecessary additional examination of incidental thyroid nodules and to systematically evaluate and report the findings of thyroid nodules. The diagnostic performance of the five representative RSSs included in this study was presented in several studies. Analysis results of the present study revealed the highest AUCs of the ACR-TIRADS, EU-TIRADS, Kwak-TIRADS, K-TIRADS, and ATA classification of 0.882, 0.843, 0.929, 0.904, and 0.859, respectively, showing high diagnostic accuracy.

ROC is an integrated result showing the performance of a diagnostic test at various thresholds by using sensitivity and specificity [87]. The value showing the highest AUC can be used as a cutoff value for diagnosis. However, AUC alone cannot draw definitive conclusions about the cutoff value. Because AUC is the result of measuring performance for all thresholds, it includes both clinically meaningful and nonsignificant values [79]. Therefore, for AUC to be clinically meaningful, it must be understood in terms of gains and losses for individual patients. Higher false-positives lead to complications and increased costs due to unnecessary additional tests, and higher false-negatives increase mortality due to disease [88]. Low sensitivity in thyroid US means that cancer may be missed and treatment may be delayed, whereas low specificity means that many unnecessary biopsies could be performed. Therefore, the cutoff value of RSS should be evaluated by the clinician on a case-by-case basis by considering both specificity and sensitivity. In other words, for patients with thyroid cancer risk factors, a high sensitivity value can be selected as a cutoff value, and in situations where overdiagnosis is concerned, it is acceptable to select a value with high specificity and AUC rather than sensitivity. On the other hand, a new thyroid ultrasound technology image such as elastography, which reflects tissue deformation when an external force is applied to the thyroid nodule, has recently been used for thyroid nodule diagnosis along with conventional ultrasound findings [89]. More clinical studies are expected to be reported on additional diagnostic methods to increase sensitivity and specificity for thyroid nodules.

This study had several limitations. First, individual characteristics of patients at high risk of developing thyroid cancer, such as sex and age, were not considered. In addition, the countries and health care facility levels in which US was conducted were not considered. Secondly, the size of the thyroid nodule was not considered. Nodule size is an important factor in follow-up and treatment decisions for thyroid nodules. Recent studies have suggested that criteria for the size of the nodule to be biopsied should be raised to avoid unnecessary procedures [88]. In 2021, modified K-TIRADS with revised biopsy criteria was proposed [90]. It was reported that modified K-TIRADS significantly reduced the unnecessary biopsy rate for small (≤2 cm) nodules while maintaining high sensitivity [27]. Therefore, when long-term clinical results are obtained for small nodules, an additional integrated cutoff value can be confirmed. Thirdly, studies were primarily retrospective in design. It seems that many prospective studies are needed for more accurate verification.

## 5. Conclusions

In this study, valid diagnostic accuracy for each RSS was confirmed, but superiority among RSS was not verified. The study confirmed the sensitivity and specificity change for each cutoff value and explained that the cutoff value can be set based on the clinical situation. When applying RSS to actual clinical practice, the pros and cons should be judged between additional examination and follow-up, with consideration of patient characteristics such as age and sex and based on the diagnostic accuracy of each cut-off value assessed.

## Figures and Tables

**Figure 1 cancers-15-00424-f001:**
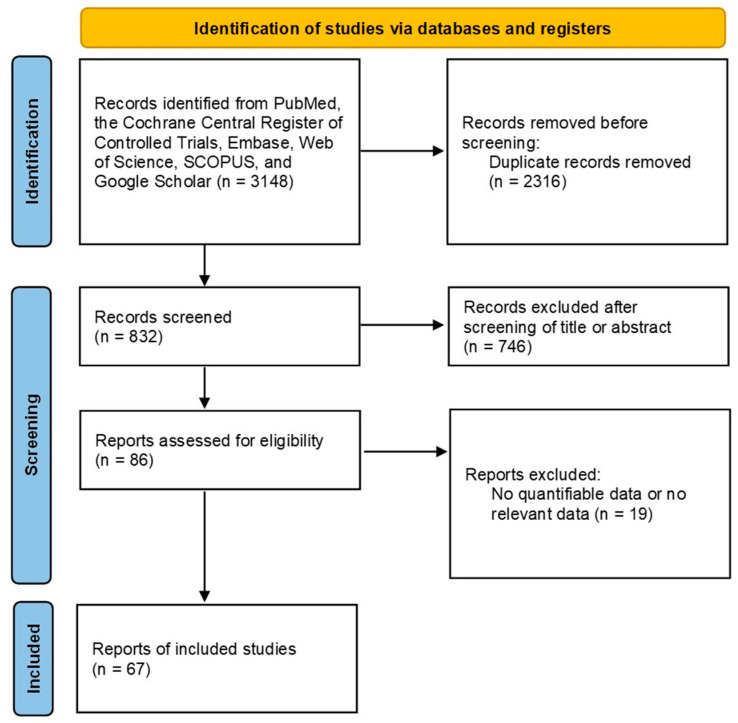
Flowchart of the study selection process for meta-analysis.

**Figure 2 cancers-15-00424-f002:**
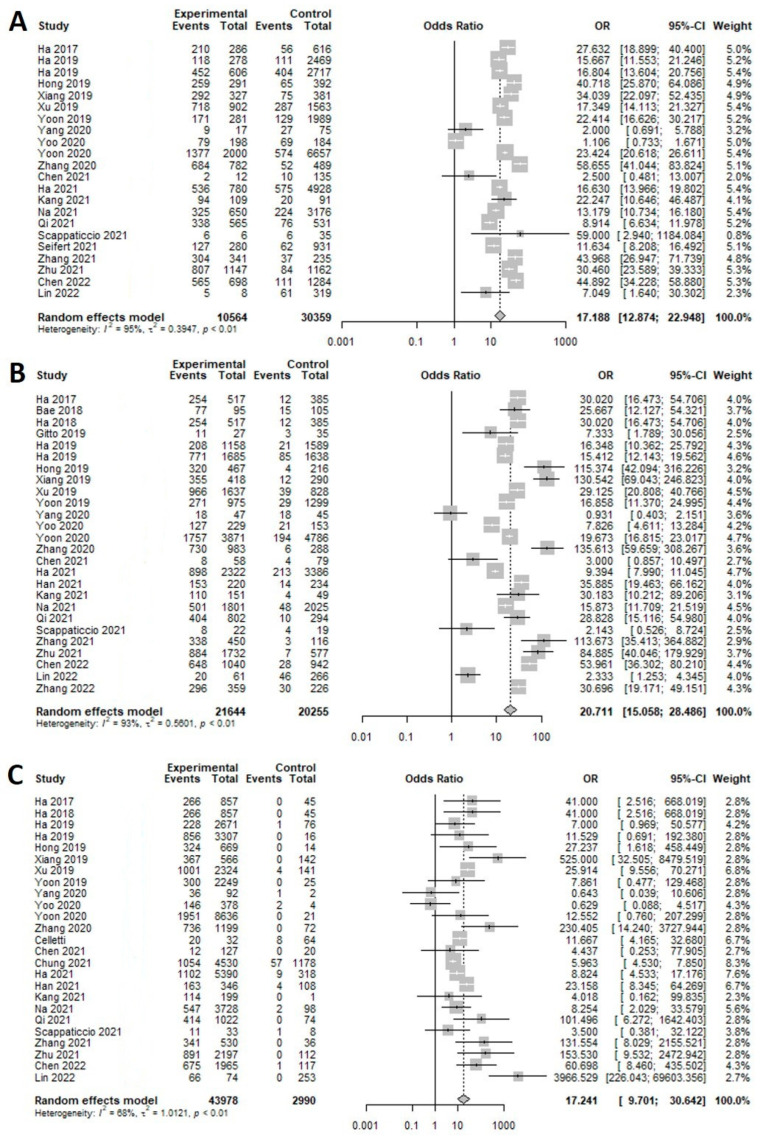
Forest plot of the diagnostic odds ratio for K-TIRAD. (**A**) High (K-TIRADS 5), (**B**) intermediate (K-TIRADS 4), (**C**) low (K-TIRADS 3).

**Figure 3 cancers-15-00424-f003:**
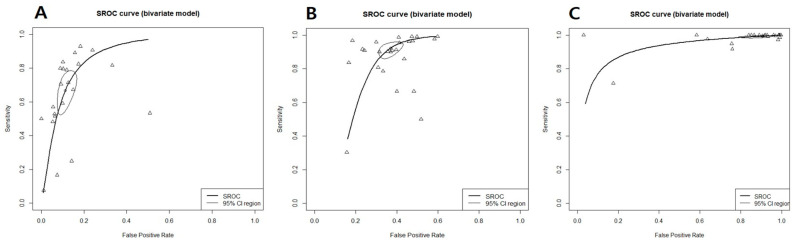
Summary receiver operating characteristic curve for K-TIRAD. (**A**) High (K-TIRADS 5), (**B**) intermediate (K-TIRADS 4), (**C**) low (K-TIRADS 3), thick curved line: summary receiver operating characteristic curve; thin circular line: 95% confident region; small circle: summary estimate; triangle: observed data.

**Table 1 cancers-15-00424-t001:** Diagnostic efficacy and the ROC curves of K-TIRADS categories.

	Sensitivity [95% CIs]	Specificity [95% CIs]	DOR [95% CIs]	AUC
High (K-TIRADS 5)	0.6644 [0.5488; 0.7632]; I^2^ = 99.1%	0.8904 [0.8495; 0.9212]; I^2^ = 98.8%	17.1881 [12.8739; 22.9479]; I^2^ = 94.7%	0.881
Intermediate (K-TIRADS 4)	0.9251 [0.8783; 0.9548]; I^2^ = 97.9%	0.6280 [0.5790; 0.6746]; I^2^ = 98.5%	20.7111 [15.0584; 28.4856]; I^2^ = 92.6%	0.792
Low (K-TIRADS 3)	0.9991 [0.9955; 0.9998]; I^2^ = 94.9%	0.0823 [0.0381; 0.1685]; I^2^ = 99.7%	17.2411 [9.7008; 30.6424]; I^2^ = 68.3%	0.904

ROC: receiver operating characteristic; CI: confidence interval; AUC: area under the curve; K-TIRADS: Korean Thyroid Imaging Reporting and Data System.

**Table 2 cancers-15-00424-t002:** Diagnostic efficacy and the ROC curves of ACR-TIRADS categories.

	Sensitivity [95% CIs]	Specificity [95% CIs]	DOR [95% CIs]	AUC
TR5 (Suspicious)	0.6350 [0.5309; 0.7279]; I^2^ = 99.2%	0.8955 [0.8613; 0.9221]; I^2^ = 98.7%	16.8442 [13.5328; 20.9658]; I^2^ = 92.5%	0.882
TR4 (Moderately)	0.9249 [0.8808; 0.9535]; I^2^ = 98.0%	0.5343 [0.4782; 0.5896]; I^2^ = 98.9%	13.6381 [9.9396; 18.7128]; I^2^ = 93.6%	0.753
TR3 (Mildly)	0.9843 [0.9698; 0.9919]; I^2^ = 96.9%	0.2289 [0.1697; 0.3012]; I^2^ = 99.5%	13.2478 [9.1596; 19.1605]; I^2^ = 85.9%	0.769

ROC: receiver operating characteristic, CI: confidence interval, AUC: area under the curve; ACR-TIRADS: American College of Radiology-Thyroid Imaging Reporting and Data System.

**Table 3 cancers-15-00424-t003:** Diagnostic efficacy and the ROC curves of ATA categories.

	Sensitivity [95% CIs]	Specificity [95% CIs]	DOR [95% CIs]	AUC
High	0.6977 [0.5992; 0.7809]; I^2^ = 98.8%	0.8715 [0.8082; 0.9161]; I^2^ = 99.4%	15.7398 [11.5605; 21.4299]; I^2^ = 95.2%	0.859
Intermediate	0.8800 [0.8239; 0.9199]; I^2^ = 97.9%	0.6155 [0.5471; 0.6796]; I^2^ = 99.2%	11.5148 [8.2698; 16.0332]; I^2^ = 95.0%	0.799
Low	0.9768 [0.9498; 0.9895]; I^2^ = 98.1%	0.2261 [0.1614; 0.3073]; I^2^ = 99.5%	6.7781 [4.1264; 11.1339]; I^2^ = 94.2%	0.694

ROC: receiver operating characteristic; CI: confidence interval; AUC: area under the curve; ATA: American Thyroid Association.

**Table 4 cancers-15-00424-t004:** Diagnostic efficacy and the ROC curves of EU-TIRADS categories.

	Sensitivity [95% CIs]	Specificity [95% CIs]	DOR [95% CIs]	AUC
High (EU-TIRADS 5)	0.7060 [0.6034; 0.7912]; I^2^ = 98.1%	0.8392 [0.7707; 0.8901]; I^2^ = 99.3%	12.2986 [9.0027; 16.8010]; I^2^ = 93.6%	0.843
Intermediate (EU-TIRADS 4)	0.9304 [0.8968; 0.9536]; I^2^ = 94.2%	0.5061 [0.4274; 0.5845]; I^2^ = 99.2%	13.0061 [9.2913; 18.2062]; I^2^ = 88.9%	0.819
Low (EU-TIRADS 3)	0.9914 [0.9763; 0.9969]; I^2^ = 91.8%	0.0303 [0.0112; 0.0795]; I^2^ = 99.3%	2.9158 [1.4936; 5.6922]; I^2^ = 74.8%	0.734

ROC: receiver operating characteristic; CI: confidence interval; AUC: area under the curve; EU-TIRADS: European Thyroid Imaging Reporting and Data System.

**Table 5 cancers-15-00424-t005:** Diagnostic efficacy and the ROC curves of Kwak-TIRADS categories.

	Sensitivity [95% CIs]	Specificity [95% CIs]	DOR [95% CIs]	AUC
5	0.1433 [0.1099; 0.1848]; I^2^ = 94.7%	0.9961 [0.9908; 0.9983]; I^2^ = 91.7%	25.8479 [12.8192; 52.1181]; I^2^ = 87.7%	0.647
4c	0.7538 [0.6426; 0.8391]; I^2^ = 98.5%	0.8904 [0.8205; 0.9352]; I^2^ = 99.2%	24.2039 [15.0245; 38.9914]; I^2^= 96.6%	0.895
4b	0.9584 [0.9308; 0.9753]; I^2^ = 94.5%	0.6379 [0.4983; 0.7575]; I^2^ = 99.5%	38.0578 [22.2904; 64.9785]; I^2^= 93.7%	0.929
4a	0.9908 [0.9799; 0.9958]; I^2^ = 95.3%	0.3286 [0.1986; 0.4914]; I^2^ = 99.8%	45.6067 [26.6992; 77.9037]; I^2^= 88.4%	0.925

ROC: receiver operating characteristic; CI: confidence interval; AUC: area under the curve; Kwak-TIRADS: Kwak-Thyroid Imaging Reporting and Data System.

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
