# Peer review of "Diagnostic Performance of Various Ultrasound Risk Stratification Systems for Benign and Malignant Thyroid Nodules: A Meta-Analysis"

_cancers, 2023, doi:10.3390/cancers15020424_

Round 1

Reviewer 1 Report

The authors describe that they evaluated the diagnostic performance of ultrasound risk stratification systems for the discrimination of benign and malignant thyroid nodules and confirmed the optimal cutoff values of individual risk stratification systems. The results implicate all ultrasound-based risk stratification systems had good diagnostic performance. The statistical analyses used are appropriate. The conclusions derived from the analyses and the interpretations are consistent.

Comments are as follows.

1. The authors should highlight the novelty of this study more clearly in the discussion.

2. It may be appropriate to include the explanation and references regarding the low-risk small PTCs (P8, Line 251).

3. In P8 Line 273, can the authors comment about the effect of other new ultrasound techiniques such as the elastography.

Reviewer 2 Report

Dear authors,

I have reviewed the manuscript "Diagnostic Performance of Various Ultrasound Risk Stratification Systems for Benign and Malignant Thyroid Nodules: A Meta-Analysis".  The manuscript is very well-written and cohesive.  I have the following comments:

1. Databases including Google Scholar were used for identifying potential references according to PRISMA guidelines. However, the method through which records were identified are not clear.  The authors presented a list of search terms (e.g., thyroid nodule), but searching "thyroid nodule" (with quotes) on Google Scholar yields over 46,000 records.  I was not able to replicate the identification of 3,148 records.  Therefore, the search methodology and record identification must be more accurately described such that the search itself is repeatable (e.g., what actual search terms were used, what filters were applied, and how many records were identified).  It is recommended that the authors read the PRISMA statement and guidelines, and follow the checklist: https://www.prisma-statement.org/documents/PRISMA_2020_checklist.docx.  Specifically, Item #7, the "full search strategies ... including any filters" must be addressed.   

2. It may be helpful for the readers if the authors included a brief description or comparison of the technical aspects pertaining to the different US RSS, including a plot of the number of articles published per year for each RSS.  This will help the readers understand the state-of-the-art and the trends/utility of the RSS.

3. It is unclear why forest plots and SROC curves for only K-TIRADS is presented in the main document.  For facile comparison, data for all RSS should be presented together.

4. Begg's funnel plots for the various studies indicate a significant amount of bias due to the existence of many data points existing outside the 95% CI lines.  The authors must discuss the impact of potential bias or explain the reason(s) why the data is like this.  The authors should include a brief description on how the funnel plots were created (e.g., how the data points were calculated, what the axes represent).
